# A deep learning framework deploying segment anything to detect pan-cancer mitotic figures from haematoxylin and eosin-stained slides

Zhuoyan Shen [1] ✉, Mikaël Simard [1], Douglas Brand [1,2], Vanghelita Andrei[3,4], Ali Al-Khader [3,4], Fatine Oumlil [4], Katherine Trevers [3,4], Thomas Butters[3], Simon Haefliger [3,5], Eleanna Kara [6], Fernanda Amary [3,4], Roberto Tirabosco[3,4], Paul Cool [7,8], Gary Royle[1], Maria A. Hawkins [1,2], Adrienne M. Flanagan [3,4,9] & Charles-Antoine Collins-Fekete[1,9]

Mitotic activity is an important feature for grading several cancer types. However, counting mitotic figures (cells in division) is a time-consuming and laborious task prone to inter-observer variation. Inaccurate recognition of MFs can lead to incorrect grading and hence potential suboptimal treatment. This study presents an artificial intelligence-based approach to detect mitotic figures in digitised whole-slide images stained with haematoxylin and eosin. Advances in this area are hampered by the small size and variety of datasets available. To address this, we create the largest dataset of mitotic figures (N = 74,620), combining an in-house dataset of soft tissue tumours with five open-source datasets. We then employ a two-stage framework, named the Optimised Mitoses Generator Network (OMG-Net), to identify mitotic figures. This framework first deploys the Segment Anything Model to automatically outline cells, followed by an adapted ResNet18 that distinguishes mitotic figures. OMG-Net achieves an F1 score of 0.84 in detecting pan-cancer mitotic figures, including human breast carcinoma, neuroendocrine tumours, and melanoma. It outperforms previous state-of-the-art models in hold-out test sets. To summarise, our study introduces a generalisable data creation and curation pipeline and a high-performance detection model, which can largely contribute to the field of computer-aided mitotic figure detection.

Mitotic activity is a crucial indicator of cellular proliferation and plays a pivotal role in cancer diagnosis and guiding clinical management[1]. Counting mitotic figures (MFs) from haematoxylin and eosin (H&E)-stained whole slide images (WSIs) is a fundamental task in pathology, required for the grading of some tumours. By convention, in clinical practice, mitotic counts are performed in the 10 most mitotically active high-power microscopic fields (HPFs) within a tumour[2]. As this is a time-consuming task, and subject to significant inter-observer variability[3–5], there has been considerable interest and effort in the development of

automated MF detection models, e.g. ICPR[6,7] and TUPAC[8,9] initiated the development of breast cancer MF datasets. Initially, mitotic detection models focused on learning handcrafted features[10–12], but recently transitioned to deep-learning-based methods that show promise[13–16]. However, MF detection remains a challenging task[17], due to the different appearance of MF in the four phases of mitosis, the range of features exhibited by abnormal MFs, as well as structures that mimic MFs (mitotic-like figures, MLFs). The above challenges are compounded by the histological heterogeneity in normal tissues and tumour types, staining

[1]Department of Medical Physics and Biomedical Engineering, University College London, London, UK. [2]Department of Radiotherapy, University College London Hospitals NHS Foundation Trust, London, UK. [3]Research Department of Pathology, University College London Cancer Institute, London, UK. [4]Cellular and Molecular Pathology, Royal National Orthopaedic Hospital NHS Foundation Trust, Middlesex, UK. [5]Institute of Medical Genetics and Pathology, University Hospital Basel, University of Basel, Basel, CH, Switzerland. [6]Department of Neurology, Rutgers Biomedical and Health Sciences, Rutgers University, NJ, USA. [7]Department of Orthopaedics, The Robert Jones and Agnes Hunt Orthopaedic Hospital, Oswestry, UK. [8]School of Medicine, Keele University, Newcastle, UK. [9]These authors contributed equally: Adrienne M. Flanagan, Charles-Antoine Collins-Fekete. ✉e-mail: zhuoyan.shen.18@ucl.ac.uk

variation between labs and differences in digital scanners used to generate WSIs.

To improve the detection of MF, the MItosis DOmain Generalisation (MIDOG)[18,19] published an updated version of their multi-domain dataset, MIDOG++[17]. This contains 503 annotated images across seven different cancer types, representing the largest currently available published dataset of MFs. The data utilised in the MIDOG studies contains the HPFs manually selected by pathologists to mimic clinical practice. However, the pathologist-led decisions may not be reproducible because of the recognised inter-observer variation[20,21], and discrepancies can be caused by the selection of areas with the densest mitotic activity[22]. In contrast, the CMC[23] and CCMCT[24] datasets used AI-assisted annotations to generate large-scale WSI datasets for MFs using canine cancers. The former used 21 WSIs of canine mammary carcinomas whereas the CCMCT dataset included 32 WSIs of canine mast cell tumours. These studies demonstrated that annotating MFs on a WSI improves the robustness of classifiers by removing the HPF selection bias and leads to a significantly higher number of detected mitoses, helping to refine further training[24].

Developing and validating pan-cancer MF detection models remains a significant challenge due to the absence of extensive pan-cancer MF datasets. One preferred approach to take forward this field of MF detection would have been to increase the size of the existing datasets incorporating multiple scanner types, staining differences across multiple sites, and tumour types. However, the lack of standardisation in the annotation protocol across various existing datasets limits their integration. For example, in the ICPR, each pixel within MFs was labelled, whereas the TUPAC only encircled MFs. MIDOG++, CCMCT and CMC utilised bounding boxes to denote the targets. We therefore took the approach to standardise the annotations by contouring nuclei of MFs. Another current limitation of this field is the lack of datasets for rare diseases. The mitotic index is important for soft tissue tumour (STT) diagnosis. Although STT represents a rare tumour group, it comprises over 100 subtypes exhibiting a wide variety of histological appearances, which can mimic other tumours, including common cancers such as melanoma, carcinoma and lymphoma. STT harbours a variable number of MFs and aids in reaching a diagnosis and predicting disease behaviour[25]. To the best of our knowledge, no publicly accessible data has been published for MFs in human STT. In this study, we contributed a large MF dataset for human STT.

Historically, targets in cellular object detection tasks are denoted using bounding boxes. However, several studies have reported that incorporating a target's mask facilitates model training and improves the overall classification performance. For instance, the Mask-RCNN outperformed the Faster-RCNN in a variety of object detection tasks[26], including MF detection[14]. The advantages of integrating nuclei contours for detection include enhancing the definition of nuclei boundaries, mitigating the morphological variability of the MFs[27] and reducing the impact of tumour histological heterogeneity. Given the constraints of a small dataset and the significant variability between mitotic cells, introducing a recognisable mitotic feature into the model aids in stabilising the training process and leads to a faster convergence.

The aim of this study was to improve the detection of MF across multiple tumour types. First, we established a large uniform database of pan-cancer MFs by deploying the Segment Anything Model (SAM)[28], a foundation object detection model, in five open-source datasets (ICPR, TUPAC, CCMCT, CMC, MIDOG++) using a single nuclei mask format. Manual revision of the masks was performed to maximise database quality. Then, we contributed an in-house dataset of human STT MFs ($N = 8400$) (Soft-Tissue MFs, STMF). The STMF was initiated by staining WSIs with an anti-phosphorylated histone H3 (pHH3) antibody to target MFs which was expanded and improved by AI-assisted annotations made by pathologists. Figure 1 illustrates the data generation pipeline for the in-house dataset, STMF, and the curation process for the multi-source datasets. The second

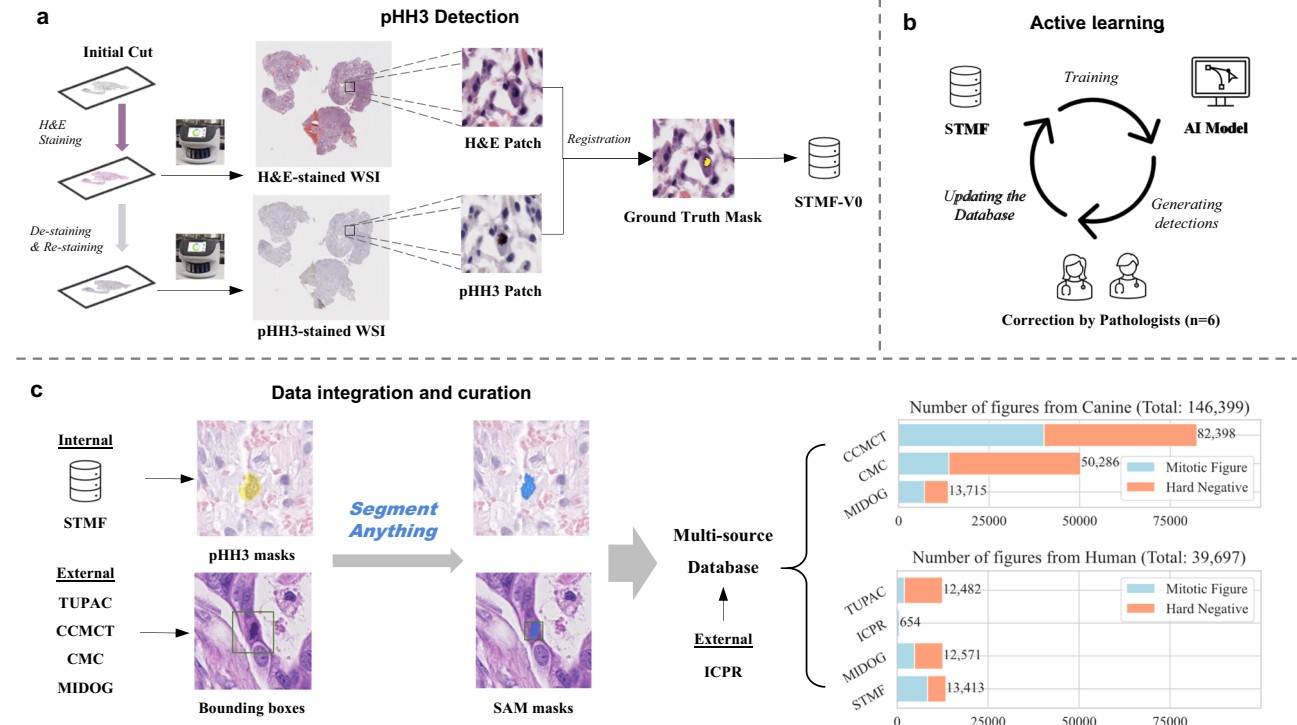

Fig. 1 | Data preparation workflow. a Haematoxylin and eosin (H&E)-stained whole slide images (WSIs) were de-stained after which immunohistochemistry was performed using an anti-phosphorylated histone H3 (pHH3) antibody which labels mitotic figures (MFs) (STMF-V0). b An initial Mask-RCNN model trained on STMF-V0 was applied to new WSIs for detecting MFs, which were then labelled by six pathologists as MF or false positives. This process facilitated the iterative refinement and expansion of the dataset to produce STMF. c The masks of the MFs from STMF and the bounding boxes from four external datasets were refined by Segment Anything (SAM) and integrated with ICPR to create the final dataset. The original and refined masks are presented in yellow and blue, respectively.

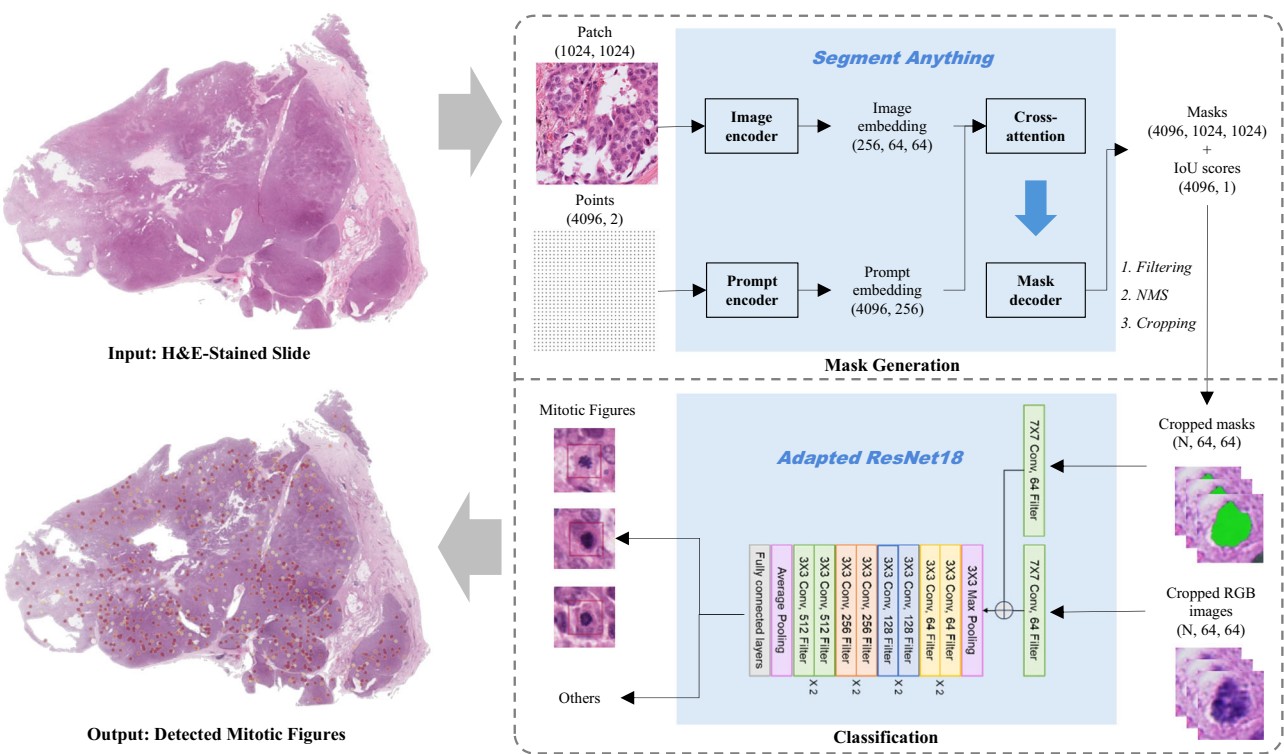

**Fig. 2 | The architecture of the OMG-Net.** The two-step architecture includes mask generation and mitotic figures (MF) classification. First, the post-process cell masks from patched WSIs are generated by Segment Anything (SAM) using an evenly sampled point grid as a prompt. Second, the RGB image of the segmented cell and the binary mask (presented in green) are used to classify MFs by employing an adapted ResNet18.

## Table 1 | Number of different types of objects in the integrated dataset

| Dataset | Tumour types | Number of images | Number of MFs | Number of MLFs | Non-MF objects |
|---|---|---|---|---|---|
| ICPR | Breast carcinoma | 100 | 654 | 0 | 10,696 |
| TUPAC | Breast carcinoma | 73 | 1999 | 10,483 | 233,992 |
| MIDOG++ | Breast carcinoma<br>Lung carcinoma[a]<br>Lymphosarcoma[a]<br>Neuroendocrine tumour[a]<br>Mast cell tumour[a]<br>Melanoma<br>Soft tissue sarcoma[a] | 392 | 9470 | 11,433 | 559,827 |
| CMC | Breast carcinoma[a] | 21[b] | 13,907 | 36,379 | 2,428,456 |
| CCMCT | Mast cell tumour[a] | 32[b] | 40,190 | 42,208 | 1,082,776 |
| STMF | Soft tissue tumour | 103[b,c] + 226[d] | 8400 | 5035 | 395,670 |
| Total | | 938 | 74,620 | 105,538 | 4,701,417 |

[a]Canine Specimens.
[b]WSIs rather than selected regions.
[c]pHH3-immunohistochemistry was used for identifying MFs.
[d]Active learning was used for annotating MFs.

objective was to develop an improved MF detection framework, which we named Optimised Mitoses Generator Network (OMG-Net). The structure of OMG-Net is outlined in Fig. 2. By integrating nuclei masks into the pre-trained classifier via a first-layer addition, we allow the model to focus on the morphological features of MFs. We demonstrate that OMG-Net is both more sensitive and specific at detecting objects, including MFs, throughout the input WSI, compared to previous models.

## Results and discussion
### Developing a large-scale MF dataset
We established a large in-house dataset for MFs in STT and merged it with five open-source datasets for MFs from human and canine specimens (Table 1). The final dataset contains 74,620 MFs and 105,538 MLFs from 712 different images or WSIs with the SAM-delineated masks for nuclei. Masks of human MFs were reviewed and modified to ensure the quality of nuclei contours. Additionally, the dataset included a large number of SAM-segmented objects, comprising tumour cells, immune cells, red blood cells, artefacts and any objects at the cell scale, collected during the data curation.

Large-scale datasets are crucial for developing AI models capable of detecting MFs effectively in a variety of cancer types and overcoming the challenges posed by the heterogeneity of staining and scanning protocols. Here, we propose a workflow for creating a reliable MF dataset:

**Table 2 | Precision, recall and F1 scores in MIDOG++ testing set of OMG-Net against the model presented by MIDOG++**

| Tumour types | Precision | Recall | F1 | Ensemble F1 | F1(MIDOG++) |
|---|---|---|---|---|---|
| Breast carcinoma | 0.82 ± 0.02 | 0.88 ± 0.02 | 0.85 ± 0.02 | 0.87 | 0.71 ± 0.02 |
| Neuroendocrine tumour | 0.64 ± 0.02 | 0.65 ± 0.03 | 0.64 ± 0.02 | 0.67 | 0.59 ± 0.02 |
| Melanoma | 0.83 ± 0.02 | 0.84 ± 0.03 | 0.83 ± 0.01 | 0.85 | 0.81 ± 0.02 |
| Lung carcinoma[a] | 0.69 ± 0.02 | 0.70 ± 0.02 | 0.69 ± 0.02 | 0.74 | 0.68 ± 0.02 |
| Lymphosarcoma[a] | 0.76 ± 0.03 | 0.74 ± 0.01 | 0.76 ± 0.03 | 0.80 | 0.73 ± 0.01 |
| Cutaneous mast cell tumour[a] | 0.84 ± 0.02 | 0.88 ± 0.02 | 0.86 ± 0.01 | 0.87 | 0.82 ± 0.01 |
| Soft tissue sarcoma[a] | 0.74 ± 0.02 | 0.73 ± 0.02 | 0.74 ± 0.02 | 0.77 | 0.69 ± 0.01 |

[a]Canine Specimens.

1. H&E destaining and employing immunohistochemistry for enhanced detection: efficient generation of a large-scale image dataset with accurate labels by detecting a substantial number of MFs on WSIs.
2. Continuous data curation: improve data quality by employing SAM to delineate precisely MF nuclei, followed by meticulous manual refinement of the generated contours.
3. Active learning: iteratively train and refine the model using a pathologist-in-the-loop approach, enabling efficient review of detected MFs, and incorporating mitotic-like figures (MLFs) and non-mitotic objects into the database for enhanced model performance.

These steps are required as it is not feasible for pathologists to annotate MFs in the numbers and the precision required by AI models, thereby affecting the diversity and size of the dataset and, consequently, the detection accuracy of the trained model. Nevertheless, each of these steps encounters limitations.

Performing immunohistochemistry following the destaining procedure of H&E-stained sections allows for the rapid and largely specific detection of MFs (specificity >99%)[29]. Still, it is not a perfect process as cells in the G2 phase of the cell cycle can exhibit weak immunoreactivity[30] as well as be prone to false-negative immunoreactivity due to the age of the slide and fixation method[31]. This restaining procedure also does not detect MLFs, which is crucial to enhance the model specificity.

Active learning can help identify MLFs but a consensus view of MF/MLF cannot always be reached by pathologists. This study highlighted the acknowledged problem of interobserver variation of MF by pathologists[4,5] which is compounded when interpreting MFs on digitised slides as it is not possible to adjust the focus plane on cells of interest. During our revision process, a notable proportion (13.8%) of AI-detected cells were categorised as 'equivocal' (Supplementary Fig. 1). A secondary review of these images performed by at least two experienced pathologists resolved some of these images but differences in opinion remained in 9.5% of AI-detected MFs.

Finally, despite the limitations discussed above, the integration of immunohistochemistry for MF detection following the destaining of H&E sections, data curation, active learning, and consensus-based review by experienced pathologists enabled us to mitigate the challenges in creating a large-scale database and developing an improved, pan-cancer MF detection model. In our future work, we will also seek to improve the training strategy to enhance the model performance when receiving data from multiple sites[32].

## Performance of MF detection in various tumours

We tested our OMG-Net on the current largest pan cancer dataset MIDOG++ containing three types of human tumours and four types of canine tumours. Table 2 shows the mean precision, recall and F1 scores with the standard deviation of the proposed framework trained five times using different random seeds, along with the F1 score obtained by ensemble voting and the F1 scores of the in-house model presented in the MIDOG++ paper.

The F1 score comparison for the three types of human tumours (breast carcinoma, neuroendocrine tumour, and melanoma) is also displayed in Fig. 3a. The MF detection scores of OMG-Net are significantly higher

($p = 0.001$) in all three types of human tumours within the testing set of MIDOG++. Figure 3b shows the benefit of combining multi-centre data, as the increased number of MFs for training correlated with an increase in the holdout F1-score.

We also compared the OMG-Net to the previous benchmarking MF detection models ranking as the top three in the MIDOG 2022 challenge[33]. Table 3 reports the scores of different models[34–36] on the subset included in both the test sets of MIDOG ++ and the MIDOG 2022 challenge (human melanoma and canine cutaneous mast cell tumour). The whole test set of the MIDOG 2022 challenge is not publicly available, and therefore we cannot compare the scores on the rest tumour types.

Object detection models such as Faster R-CNN[37], RetinaNet[38] and YOLO[39] have been widely used for MF detection[13,24]. These models integrate in a single model an object proposal network with a primary classifier. However, these models suffer from the imbalanced loss problem, as the cell segmentation and classification loss have inherently unequal magnitudes. The gradient updates that occur during back-propagation can be dominated by the loss function with the larger norm[40], leading to suboptimal training and convergence issues. This becomes even more prominent when dealing with small datasets or complex objects, as the imbalance in the loss functions' impact can significantly hinder the model's ability to learn effectively from the limited available data[41]. The use of integrated object detection models in histopathological studies has been shown to generate false positive results due to the complex and variable nature of cell morphology. Advanced object detection models are constantly evolving while their performance in MF detection has not been reported. We retrained an RT-DETR model[42] using our dataset and tested it on the same test set. The dataset was prepared as patches with a size of 640 pixels × 640 pixels. An RT-DETR-X was trained for 200 epochs using the default configuration. Overall, we observed a lower F1 score than OMG-Net (0.764 ± 0.01 vs 0.783 ± 0.02).

It has been demonstrated[43,44] that integrating a secondary classifier, trained on MFs and other objects such as MLFs, to review and reject false positive cases improves a framework's precision. This approach limits the imbalanced loss problem, as the segmentation loss is excluded in training the additional classifiers. However, these methods add unnecessary complexity to the network since two classifiers must be trained.

To mitigate the imbalanced loss problem, we elected to separate entirely the object detection and classification steps. This offers an innovative approach that differs from those previously published. Instead of training an object detection model for generating objects that are highly likely to be MFs, all the objects at the cell scale are segmented by SAM from the ROIs and classified, improving the sensitivity of our model. Other objects, including immune cells, cells not in mitosis, and artefacts generated during the data preparation stage, can also be used to train the classifier, improving its capability to reject false positives. We also compared ResNet18 to ResNet50, DenseNet121, ConvNeXt, a transformer-based advanced image classifier, and EfficientNet-B7, the classifier used by the top 1 team on the MIDOG 2022 leaderboard (Supplementary Table 1). Since the classifier is for discriminating MFs from other cells, the input size is relatively small (64 pixels × 64 pixels). Although more advanced larger models

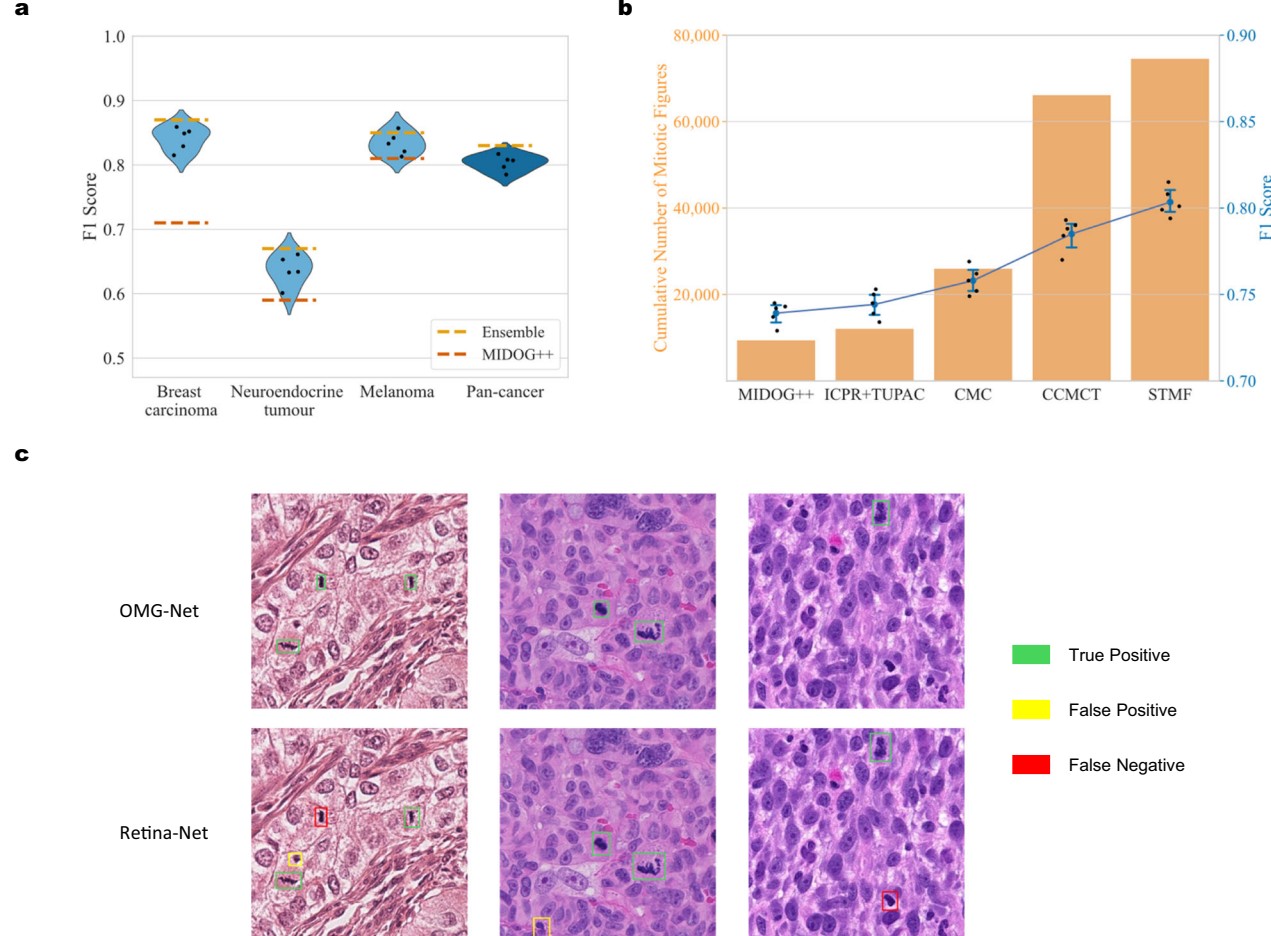

**Fig. 3 | Detection performance. a** The testing F1 scores in the human subsets of the proposed framework. The scores are presented as individual data points (*n* = 5 independent experiments), with yellow dashed lines marking the ensemble F1 scores and red dashed lines marking the mean F1 scores reported by MIDOG++. **b** The changes in the average F1 score as more mitotic figures (MFs) are included in training. The scores are presented as individual data points (*n* = 5 independent experiments), with error bars presenting the 95% confidence intervals. **c** The detection results of the OMG-Net and the Retina-Net used in the MIDOG++ in example regions. The green, yellow and red bounding boxes represent the true positives, false positives and false negatives.

**Table 3 | F1 scores of OMG-Net and the top 3 models from the MIDOG 2022 challenge in the overlap subsets between the MIDOG ++ and MIDOG 2022 test sets**

| Tumour types | OMG-Net | TIA centre[34] | TCS research[35] | USZ/UZH Zurich[36] |
|---|---|---|---|---|
| Melanoma | 0.85 | 0.80 [0.74,0.84] | 0.76 [0.66,0.80] | 0.79 [0.74,0.83] |
| Cutaneous mast cell tumour[a] | 0.87 | 0.83 [0.81,0.86] | 0.76 [0.58,0.83] | 0.73 [0.66,0.79] |

[a]Canine Specimens.

show better performance in classifying more complex and diverse images, they did not bring better performance on our task with a small input size for binary classification.

## Nuclei contours from the Segment Anything Model enhanced the detection performance

As shown in Fig. 4a, the appearance of MFs is highly diverse, exemplified by atypical MFs. The segmented mask may not fully cover the MFs or may contain background noise. To refine the training process, we reviewed the masks in the human subset of MIDOG++ (4435 in breast carcinoma, 2075 in melanoma and 2400 in neuroendocrine tumour), and adjusted the SAM prompt when required. The impact of this manual curation was assessed by comparing the F1 scores of the models only with RGB images (RGB Classifier), the score of the model with zero-shot SAM mask input (RGB-M0

Classifier) as well as the score of the model with reviewed and refined masks (RGB-M1 Classifier), with results shown in Fig. 4b.

Compared to the model without masks (RGB), the RGB-M0 model yielded higher F1 scores for detecting MFs from breast carcinoma (*p* = 0.011) and melanoma (*p* = 0.001) but not for neuroendocrine tumours. Upon further analysis, we noted that the fraction of masks requiring a second adjustment was higher in neuroendocrine tumours (16%), compared to breast carcinoma (8%) and melanoma (5%). As predicted, the RGB-M1 Classifier showed the best performance and significantly outperformed the RGB Classifier for breast carcinoma (*p* = 0.00018), melanoma (*p* = 0.00032) and neuroendocrine tumours (*p* = 0.021). We conclude that the low-quality masks, which may include surrounding backgrounds or exclude part of the nuclei (Fig. 4b), can impact the performance of the RGB-M0. Further examples of failed prompts are shown in Supplementary Fig. 2.

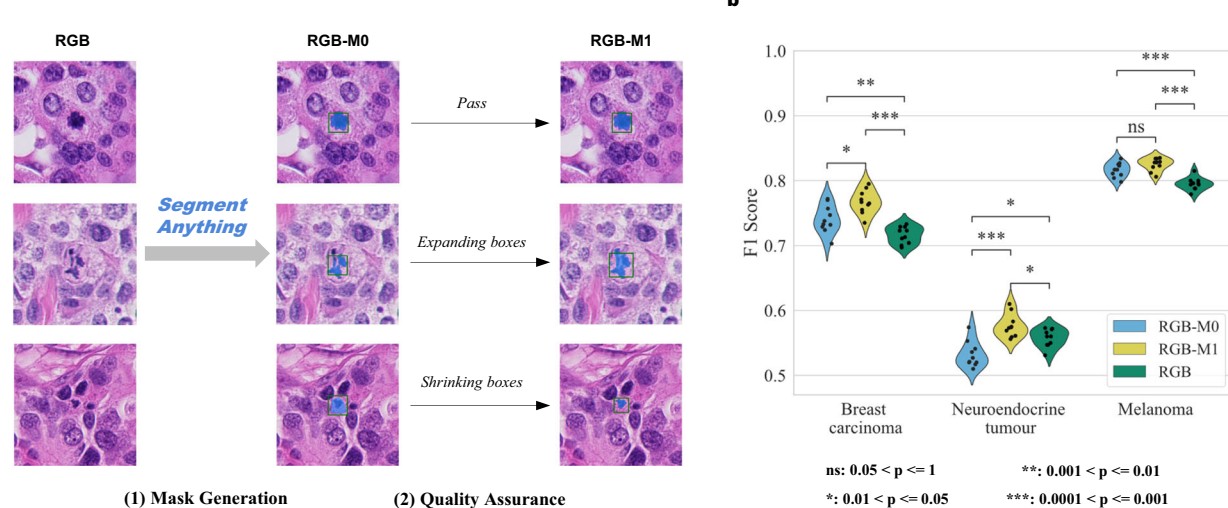

**Fig. 4 | The contribution of Segment Anything (SAM) masks. a** Illustration of the quality assurance process for the MIDOG++ human subset. **b** F1 scores of the classifier using only RGB images (RGB Classifier), the classifier using additional SAM masks (RGB-M0 Classifier), and the model using reviewed SAM masks (RGB-M1 Classifier). The scores are presented using violin plots with individual data points (*n* = 10 independent experiments).

The accuracy of our models varied considerably across different tumour types, with neuroendocrine tumours exhibiting significantly lower performance, which was consistent with the results of the MIDOG++ algorithm. In parallel, we observed a higher proportion of low-quality masks in neuroendocrine tumours (16%) compared to breast carcinoma (8%) and melanoma (5%), suggesting that the quality of the training data may have contributed to the disparities in model performance across these cancer types. Even then, manual curation of the masks helped improve significantly the model detection performance.

To decide which foundational model to select as a nuclei detector, we evaluated published fine-tuned variants of SAM against the overall mask quality for cells in histology images. Specifically, we tested the original SAM, CellSAM[45], MicroSAM[46] and CellViT[47] on the Lizard dataset[48], which is the largest pan-cancer dataset with nuclei labels. We compared the DICE of models in Supplementary Table 2 and visualised some example regions segmented in Supplementary Fig. 3. Overall, SAM achieved the highest DICE score (0.76 ± 0.13). Since the ground truth contours in the Lizard dataset are from manual annotations and inter-observer variability was observed, the imperfect DICE score does not necessarily indicate limited segmentation quality, but merely a disagreement with pathologists.

The lower score of CellSAM is likely due to it being fine-tuned on the ViT-B architecture, which has lighter weights compared to the ViT-H architecture used by OMG-Net. CellViT reported limited performance in detecting inflammatory cells and connective tissues[47], which suggests that lower DICE in general nuclei segmentation was expected. The worst performance of MicroSAM can be attributed to its lack of fine-tuning on H&E-stained images and its broader training on a variety of microscopy images. Besides, their preprint paper indicates that while the model performs well when bounding boxes are provided as prompts, the quality of segmentation decreases significantly during automatic mask generation.

Based on this analysis, we elected to keep the original SAM as the cell detector in our study. Future work will include refining the SAM object-proposal method for H&E-stained specific cell types.

### Canine mitotic figures help to train the detection of human mitotic figures

The merged and uniform dataset contains a significant proportion of canine MFs with examples from both human and canine WSI displayed in Fig. 5a. The inclusion of the canine data significantly improved the detection of MFs in breast carcinoma (*p* = 0.007) and neuroendocrine tumours (*p* = 0.015)

and the F1 score in melanoma was also marginally increased (*p* = 0.080) (Fig. 5b).

### Including mitotic-like figures and non-mitotic objects is key to improving model precision

Besides MFs, MLFs were also labelled in the original dataset. MLFs represent morphological structures that resemble MFs including pyknotic nuclei, apoptotic bodies, and neutrophil polymorph amongst others, often mis-classified as MFs. An example is displayed in Fig. 6a. Apart from MLFs and tumour cells, the SAM-curated dataset contains other cells, including immune cells, red blood cells and any objects at the cell level such as arte-facts, segmented by the SAM during data curation to present the classifier with a heterogeneous set of data. These were used in the training step to augment the original dataset and provide the model with a diverse repre-sentation of segmented objects. Figure 6b shows that the model including non-MF objects (SAM-AUG) has significantly higher precision for all three types of tumours (*p* = 0.008) compared to the model trained only with MFs and MLFs (original). As expected, the recall remains unchanged, and the overall F1 scores are improved (*p* = 0.007).

### The remaining challenges in deploying computer-aided MF detection approaches

**Detecting MFs in rare tumours.** Although a large number of studies reported MF datasets or detection models for breast cancer, detecting MFs in other tumour types, especially rare cancers remains challenging. For instance, the detection scores for neuroendocrine tumours are notably lower compared to that of other human tumours, which can be caused by several reasons:

The number of annotations for neuroendocrine tumours is relatively small (524) in the training data. All the annotations were from the MIDOG ++ and the images were scanned using the same scanner. The small amount of data and the lack of diversity make it more challenging to develop robust and accurate models.

We will keep working on increasing the size of the training datasets by constantly collecting specimens from RNOH and collaborative institutions and applying the data generation and curation pipeline to expand the size and diversity of neuroendocrine tumours. By incorporating a larger and more diverse set of annotated images, the model will be accurate and robust in detecting MFs from neuroendocrine tumours.

1. The poor performance can also be attributed to the biological char-acteristics of neuroendocrine tumours. Cell aggregation, where cells

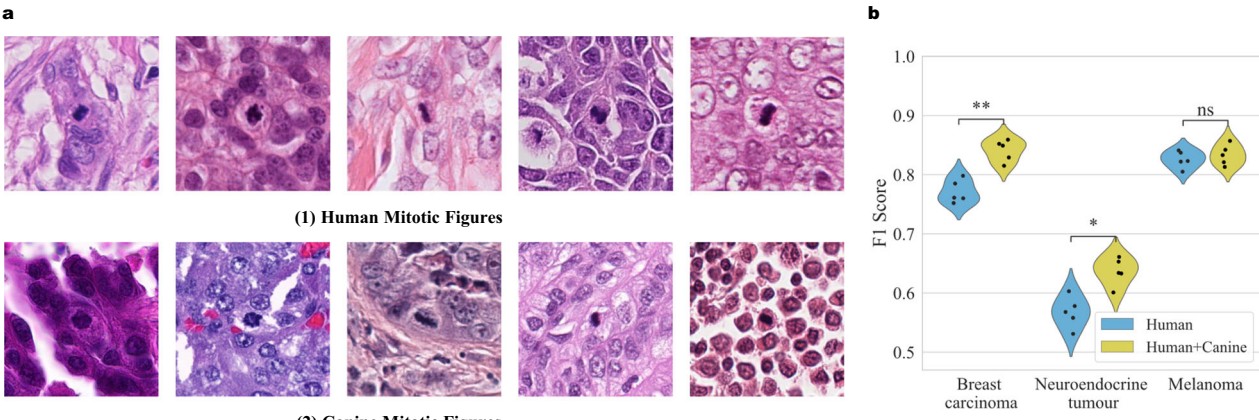

**Fig. 5 | Including the canine mitotic figures (MFs) for training improves the detection. a** Example of MFs in human and canine haematoxylin and eosin (H&E)-stained sections. **b** The F1 scores of the models trained with only human data and with both human and canine data. The scores are presented using violin plots with individual data points (*n* = 5 independent experiments).

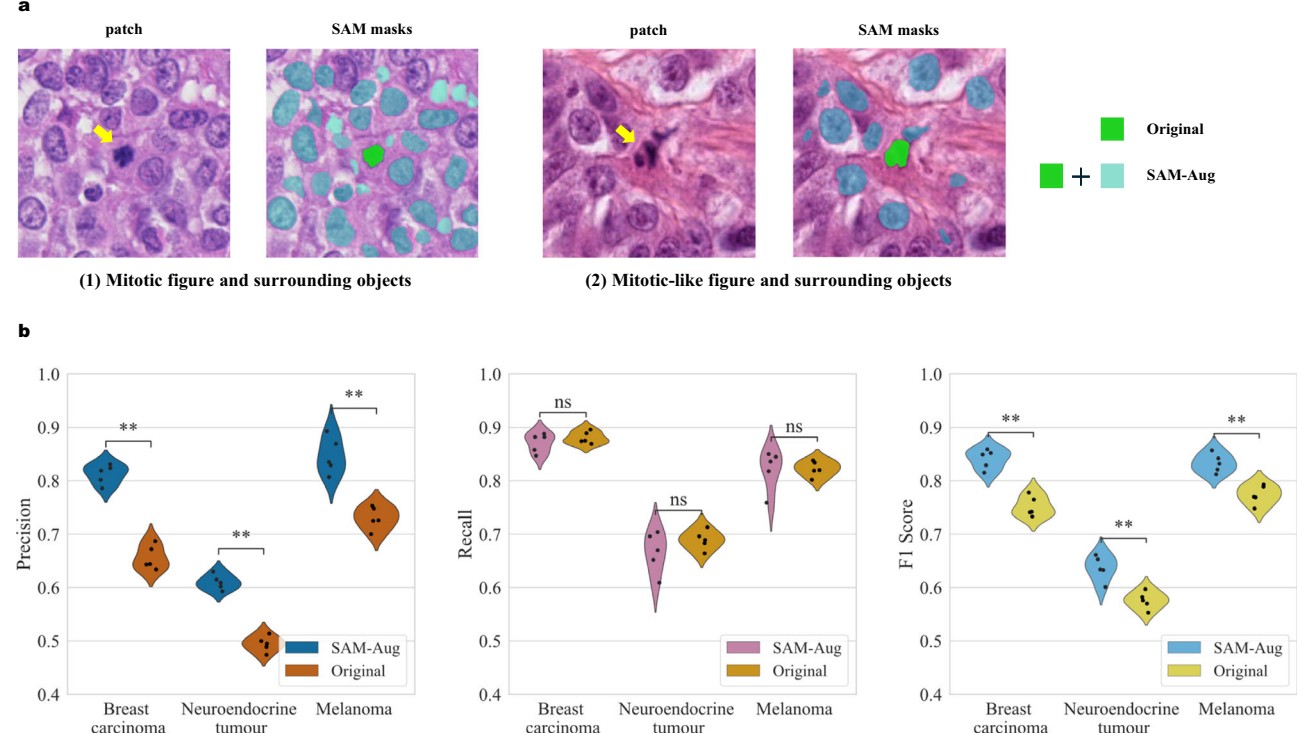

**Fig. 6 | Including mitotic-like figures (MLFs) and non-mitotic objects for training improves the detection. a** Example of patches containing a mitotic figure (MF) (left) and an MLF (right). The MFs and MLFs are masked in green (Original data). The surrounding cellular components segmented by Segment Anything (SAM) are marked in light blue and are added to the MFs and MLFs (SAM-Aug data). **b** The precision, recall and F1 scores of the model trained with the Original data and the model trained with SAM-Aug data. The scores are presented using violin plots with individual data points (*n* = 5 independent experiments).

clump together, is often observed in neuroendocrine tumour specimens, for which it is difficult to distinguish individual cells' boundaries and limit the quality of the nuclei masks. The heterogeneity of aggregation also brings more diverse cell morphology, making it more challenging to recognise and classify them correctly.

2. In our dataset, the images of neuroendocrine tumours frequently suffer from poor quality, including issues like blurring, uneven staining, and artefacts. Low-quality images can lead to increased false negatives or false positives, as the model struggles to interpret the compromised visual information.

To improve the quality of the WSIs used for training, standardised protocols for slide preparation, staining and scanning will be deployed in our future work. Meanwhile, we will implement quality control by pathologists before selecting images for developing the model.

The limited number of cases but high degree of heterogeneity of rare cancers such as STT and neuroendocrine tumours, complicates the acquisition of ample and diverse data for training robust AI models. In this study, we established the largest-to-date MF dataset for human STT. We are committed to continuously expanding this dataset and refining our models to enhance their accuracy and utility in sarcoma diagnosis. Meanwhile, we

will also attempt to use semi-supervised learning approaches[49,50] to tackle the challenge of insufficient size and diversity of data for rare subtypes.

**Comparing the mitotic index identified by AI and by pathologists.** Clinically, the mitotic index is obtained by pathologists counting the number of MFs within 10 HPFs. However, this process is subject to intra- and inter-observer variability, which can significantly affect the consistency and reliability of the measurements, bringing challenges to aligning manual counts to the mitotic index given by AI-driven WSI analysis. Standardised protocol is needed for comprehensively assessing the accuracy and potential clinical impact of the AI mitotic index.

**Clinical implementation of AI models for mitotic figure detection.** While AI-based models for detecting MFs have demonstrated exceptional performance in research settings, translating these advances into clinical practice remains a challenge. The key issues include allocating sufficient computational resources in the clinic, ensuring compliance with clinical data regulation and receiving feedback and new data for constant model improvement. Our future work also includes designing a federated learning platform to facilitate the model refining, which allows multiple institutions to expand the training data while keeping sensitive data localised. We also aim to deploy reinforcement learning by including pathologists to constantly correct the detection and give tumour type-specific and image-specific feedback to enhance the model performance.

In conclusion, we have established a large-scale MF dataset by integrating five open-source datasets acquired from multiple centres including an in-house dataset of STT. Using the SAM-enhanced dataset, we employed a novel two-step framework, OMG-Net, where SAM served as the object detector followed by an adapted ResNet18 as the MF classifier. This approach improved the accuracy of MF detection from various human tumours including breast carcinoma, neuroendocrine tumours and melanoma compared to existing state-of-the-art models. Future steps include a head-to-head prospective assessment of this model with pathologists' scores for MFs before introduction into safe clinical practice.

## Methods
### Dataset
**Ethical approvals.** The data involved in the STMF dataset are collected in the Royal National Orthopaedic Hospital (RNOH) NHS Trust under the Health Research Authority (HRA) and Health and Care Research Wales (HCRW) Approval. Integrated Research Application System (IRAS) project ID: 328987. Protocol number: EDGE 161548. Research Ethics Committee (REC) reference: 23/NI/0166. Informed consent was obtained from all human participants. All ethical regulations relevant to human research participants were followed.

**Open-source datasets.** We integrated five open-source datasets (ICPR, TUPAC, CCMCT, CMC, MIDOG++), comprising 68,687 MFs from eight different scanners and eight types of human and canine tumours. The types of tumours studied and scanners are listed in Supplementary Table 3. All the images were scanned in 40× magnification with a pixel size of ~0.25 μm.

**In-house dataset.** We describe a workflow for utilising an anti-pHH3 antibody to specifically detect MFs and expand the dataset by active learning (Fig. 1). The number of MFs in each diagnosis of STTs is listed in Supplementary Table 4.
- pHH3-assisted MF detection: the pHH3 antibody employed specifically detects the core protein histone H3 only when phosphorylated at serine 10 (Ser10) or serine 28 (Ser28), thereby identifying mitotic cells within a tissue sample[51]. We selected 94 archived slides and tissue blocks from STTs and prepared fresh H&E tissue sections which were then scanned for generating our dataset. These H&E-stained tissue sections were then de-stained after which immunohistochemistry was performed using a rabbit monoclonal

(RM) hybridoma Ser10 pHH3 [BC37] (Company: BIOCARE MEDICAL, Catalogue Number: ACI 3130A, C, Dilution 1:100)[30] and then counterstained with eosin. The masks of the MFs were extracted from pHH3-immunolabelled WSIs by setting thresholds for the RGB values and transferred to the same location on the matching H&E-stained WSIs. Registration between the pHH3-immunolabelled and H&E-stained WSIs was achieved by random sample consensus (RANSAC)[52] on both a WSI level and patch level. The contours and positions of 7952 MFs (STMF-V0), were identified and validated by pathologists reviewing the H&E and immunolabelled sections. However, not all mitoses were identified by pHH3-labelling indicating that the antibody was not entirely sensitive[53].
- Active Learning: Although the identification of cells in mitosis by pHH3 can establish a dataset with a large number of MFs, it cannot identify MLFs, and models trained only with IHC suffer from limited precision. Active learning is required to augment the dataset with MLFs.

During the active learning process, pathologists corrected the image labels given by a machine learning model and fed them back to re-train the initial model, so that the model performance for the target task can be continuously improved during the iteration of machine-generating and human-labelling.

To expand the STMF-V0 dataset, we trained an initial Mask-RCNN model on it and applied the model to new WSIs for detecting MFs. The AI-detected MFs were randomly assigned to six pathologists to be independently labelled as 'MF', 'not MF' if the pathologist could confidently make a decision, or 'uncertain' when the morphological features were equivocal. These equivocal MFs were reviewed by two senior pathologists. Other structures such as apoptotic bodies were also labelled to create the final dataset, STMF, with 8400 MFs and 5035 MLFs.

### Data curation
The MFs were annotated using bounding boxes in the CCMCT, CMC and MIDOG++ datasets. However, the size of the boxes varies due to the lack of standard annotation criteria. We hypothesised that the contours of nuclei could provide extra information for classifying MFs, as the model would be guided to focus on the most representative pixels of the nuclei rather than the surrounding environment.

We use the bounding boxes provided in the CCMCT, CMC and MIDOG++ datasets as prompts to generate the masks using SAM. To ensure the quality of the automatically generated masks, we inspected individual masks of the MFs from three types of human tumours in MIDOG++. The percentage of masks amended following review is 8%, 5% and 16% out of 4435 masks in breast carcinoma, 2075 in melanoma and 2400 in neuroendocrine tumour, respectively. In total, only 8% of the masks required a second inference of SAM using adjusted bounding boxes. Since the cells can be distorted during the de-staining and pHH3 labelling process, we also applied the SAM to the STMF using the outside boxes of the pHH3-immunolabelled masks as prompts. The numbers of MFs and MLFs from human and canine samples are shown in Fig. 1. Quality assurance was done for masks of all the human samples, whereas the generation of masks in canine sections was fully automated.

### OMG-Net: a two-stage detection framework
The proposed framework consists of two steps:
- The SAM was applied to patches of 1024 × 1024 pixels from the WSIs after background removal. This process is performed by analysing a grayscale low-resolution version of the WSIs to filter out background areas. Assuming the edges mostly contain background, the background threshold is estimated based on the perimeter. The tissue pixel fraction in each tile is then calculated based on the threshold to identify tiles with enough tissue content. In the non-background tiles, 64 points were evenly sampled along each dimension, totalling 4096 points used as prompts per patch. The quality of the masks was predicted by two factors, an AI-predicted Intersection over Union (AI-IoU) and a

stability score. The AI-predicted IoU comes from an adjacent multi-layer perceptron in the mask decoder section of SAM. The stability score is the IoU between the binary masks obtained by thresholding the predicted mask logits at high and low values. Only the objects with AI-IoU scores and stability scores higher than 0.8 and areas between 2.25 μm² and 225 μm² were kept after filtering. The filtered masks were then ranked by their AI-IoU scores. Non-maximum suppression[54] was used to remove duplicated masks.

- The objects generated were then classified by the second model, a ResNet18 pre-trained on ImageNet, as MFs or other objects. In addition to taking a 3-channel RGB image, the mask of the object was encoded by a convolutional layer and summed to the first convolutional layer of the ResNet18. Via this process, we retained the ability to use pre-trained models while providing extra mask information to the model.

## Model development and testing

The framework was implemented using Pytorch and Pytorch Lightning and was trained using a single NVIDIA GeForce RTX 3090 for 30 epochs with a batch size of 8000. The learning rate was set up at 0.001, optimised by the AdamW algorithm[55] and cosine annealing scheduler[56].

**Training and validation**. We trained the ResNet18 to classify MFs while the SAM mask generator was not retrained. The SAM was applied to all patches in the dataset after data curation, and the other objects surrounding the targets were also segmented and included in the training and validation data. The binary classifier is trained on two classes: (1) MFs and (2) labelled MLFs and other cells or objects segmented by SAM. In each training process, 90% of the data was used for training the model, while the remaining 10% was used for validation. The training was repeated five times using different random seeds to get five models with different data splits.

**Data augmentation**. Colour and spatial augmentation were applied to the training data to reduce the impact of the staining variation and increase the robustness of the model. To achieve colour augmentation, RGB images are deconvolved into H&E stains using the stain vectors proposed by Ruifrok and Johnston[57]. The stain concentration perturbation scheme introduced by Tellez et al.[58] was used with a uniform sampling and $\sigma = 0.14$ on the deconvolved H&E channels prior to reconstructing RGB images. Random horizontal flips ($p = 0.4$) was also used.

**Test set and performance metrics**. We used the same testing set provided by MIDOG++, which contains 2467 MFs from 105 sections of three types of human tumours and four types of canine tumours. Precision, recall and F1 score were used to evaluate the performance of our mitotic detection framework. They were calculated by

$$Precision = \frac{N_{TP}}{N_{TP} + N_{FP}}$$

$$Recall = \frac{N_{TP}}{N_{TP} + N_{FN}}$$

$$F1 = 2 \cdot \frac{Precision \cdot Recall}{Precision + Recall}$$

where $N_{TP}$, $N_{FP}$ and $N_{FN}$ represent the number of true positives, false positives and false negatives, respectively.

## Statistics and reproducibility

All statistical analyses were conducted using SciPy (v1.14.1). The Mann–Whitney U test[59] was employed to compare classification scores between two independent groups of experiments. Each experiment within the groups was performed using k-fold cross-validation with 15% data used for validation to reduce the variability associated with a single data split. The scores of each group were presented as mean and standard deviation. The number of experiments in each group and the exact *p*-values are reported in the results section. All the *p*-values are two-sided and a *p*-value of less than 0.05 was considered statistically significant.

## Reporting summary

Further information on research design is available in the Nature Portfolio Reporting Summary linked to this article.

## Data availability

All MF images and their SAM-dilated contours are available without restriction via the Zenodo repository (https://doi.org/10.5281/zenodo.14246170)[60] in accordance with the UKRI Common principles on research data. The original images and annotations of the open-source datasets can be found via their repositories: ICPR[6,7]: http://ludo17.free.fr/mitos_2012/index.html. TUPAC[8,9]: https://tupac.grand-challenge.org/. CMC[23]: https://github.com/DeepMicroscopy/MITOS_WSI_CMC. CCMCT[24]: https://github.com/DeepMicroscopy/MITOS_WSI_CCMCT. MIDOG++[18,19]: https://github.com/DeepMicroscopy/MIDOGpp. Source data for all the figures in the manuscript is provided in Supplementary Data 1. All other data are available from the corresponding author upon reasonable request.

## Code availability

The code for data generation and model implementation is provided on the GitHub repository. https://github.com/SZY1234567/OMG-Net. A fixed version of the code is also available via the Zenodo repository (https://doi.org/10.5281/zenodo.14246170)[60]. The original code for the SAM can be found via their repository: https://github.com/facebookresearch/segment-anything.

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

## Acknowledgements

This project is supported by the UKRI Future Leaders Fellowship (MR/T040785/1), EPSRC Research Grant NR1 (EP/Y020030/1), the Radiation Research Unit at the Cancer Research UK City of London Centre Award (C7893/A28990), as well Sarcoma UK (Award SUK18.2021). A.M.F. and K.T. are supported by the National Institute for Health Research, UCLH Biomedical Research Centre, and the CRUK Experimental Cancer Centre, as well as the Royal National Orthopaedic Hospital R&D Department. A.M.F. is also supported by Bone Cancer Research Trust (infrastructure) Grant, Chordoma UK, the Royal National Orthopaedic Hospital NHS Trust Charity, and the Tom Prince Trust. S.H. was funded by the Children's Cancer Foundation Basel (Grant: C23-2021-21). T.B. is a Ph.D. Clinical Fellow funded by the Jean Shanks Foundation and the Pathological Society of Great Britain and Ireland. E.K. was funded by the Royal National Orthopaedic Hospital.

## Author contributions

The study was designed by Z.S., C-.A.C-.F. and A.M.F. Z.S., C-.A.C-.F. and M.S. developed the code for the models. A.M.F., V.A., A.A.K., S.H., E.K., and T.B. provided the WSIs and annotations. F.O., K.T., F.A., and R.T. oversaw slide acquisition and anonymisation. Z.S. and C-.A.C-.F. contributed to the data generation and curation. P.C. contributed to the statistical analysis. G.R., D.B. and M.A.H. contribute to the discussion and interpretation of the results. Z.S. wrote the first draft of the manuscript. C-.A.C-.F., M.S., G.R., D.B., P.C., M.A.H. and A.M.F. contributed to writing and improving the manuscript. All authors contributed to the critical revision of the paper. The authors acknowledge Rohan Tapabrata Chakraborty for fruitful discussions.

## Competing interests

The authors declare no competing interests.
