## [Transparent Peer Review file · Communications Biology]

A Deep Learning Framework Deploying Segment Anything to Detect Pan-Cancer Mitotic Figures from Haematoxylin and Eosin-Stained Slides

Corresponding Author: Dr Zhuoyan Shen

Version 0:

Reviewer comments:

Reviewer #1

(Remarks to the Author)

This study introduces an AI-aided approach to improve the detection of mitotic figures (MFs), a crucial factor in cancer grading, in digitized haematoxylin and eosin-stained whole slide images. Recognizing MFs is traditionally time-consuming and prone to human error, which can lead to incorrect cancer grading and suboptimal treatment. To address these challenges, the researchers created the largest pan-cancer dataset of MFs, combining an in-house soft tissue tumor dataset with five open-source datasets, resulting in 74,620 MFs. They developed a two-stage framework, OMG-Net, which uses the Segment Anything Model (SAM) for cell contouring and an adapted ResNet18 for MF classification. OMG-Net achieved an F1-score of 0.84 in pan-cancer MF detection, significantly outperforming the previous state-of-the-art MIDOG++ model, especially in breast cancer detection, by 16%. This approach offers superior accuracy in detecting MFs across various tumor types and scanning methods.

Weakness:

1. The paper lacks an analysis of the disadvantages of current mitotic activity detection models.
2. The work primarily focuses on methodological aspects, with insufficient discussion on the motivation of reinforcement learning. More context on the specific topic is needed.
3. The authors seem to have missed some relevant literature. Specifically, they don't discuss learning-based methods for image-level tasks, missing out on several relevant citations: "Class-aware adversarial transformers for medical image segmentation", "Momentum contrastive voxel-wise representation learning for semi-supervised volumetric medical image segmentation", "SimCVD: Simple contrastive voxel-wise representation distillation for semi-supervised medical image segmentation", "Incremental learning meets transfer learning: Application to multi-site prostate mri segmentation", "Bootstrapping semi-supervised medical image segmentation with anatomical-aware contrastive distillation", "Unsupervised wasserstein distance guided domain adaptation for 3d multi-domain liver segmentation", "Mine your own anatomy: Revisiting medical image segmentation with extremely limited labels", "Rethinking Semi-Supervised Medical Image Segmentation: A Variance-Reduction Perspective", "ACTION++: Improving Semi-supervised Medical Image Segmentation with Adaptive Anatomical Contrast", and "Implicit Anatomical Rendering for Medical Image Segmentation with Stochastic Experts". These methods are relevant to the method proposed in this paper. These relevant papers should be included in the reference list.

Reviewer #2

(Remarks to the Author)

Very interesting paper and good work in general.

1) Line 99: From what I understand, you used the pHH3 as a marker to generate a training dataset. How was the staining protocol done? From my understanding, if you first stain with H&E and then with the IHC, due to the quite heavy processing of the tissue to apply the H&E (heating up to a high degree) and the destaining process, it leaves the tissue quite altered. Also, pHH3 can detect MF but it might produce "false positives" due to the fact that it detects the mitotic process quite early even on cells that might not appear to be mitotic figures on the H&E. Did you have some classifying the pHH3 detections as well?

2) Figure 2: When using the SAM to do the segmentation, how do you ensure that SAM will segment structures of cell level and not larger structures?

3) Line 127: You mention that you benchmarked against the current state-of-the-art MIDOG++ but the MIDOG++ is a dataset and the model presented in the paper is not state-of-the-art. It was used merely to demonstrate how the dataset varies and the detection performance when an off-the-shelf detector is used. The state-of-the-art would be the top-3 models on the leaderboard of the MIDOG grand challenge.

4) line 129: typo 'hows'  'shows'

5) Table 2: You chose to compare against a subset of the dataset using the leave-one-out(?) training results. Why was this choice made?

6) Line 146: You refer to "only RGB images" and "RGB classifier". What do you mean exactly? Is there a dataset of non-RGB images?

7) Line 224: You mention that decoupling the object detection and classifier steps improves the performance because the existing models have to reject many false positives due to the imbalance of the dataset. However in your description you use SAM to generate masks and then apply a classifier. How does that approach exactly generate less false positives? If you were to segment all cells in a ROI, it would surely generate as many false positives as a detector I would assume.

8) You used for the classifier a ResNet-18, did you test other models as well?

9) I would like to see the performance of this method using the same method as that was outlined during the grand-challenge. That means running in on the full test set, not just the human in order to have a direct comparison with the leaderboard of the challenge. By only comparing specific tumor types, it feels like cherry-picking the results. If the method is better, then it would perform better on the whole test dataset. After you have shown in direct comparison that the method is better, you can further evaluate with an extensive training dataset.

Reviewer #3

(Remarks to the Author)

This work first presents an annotated mitotic-figure (MF) dataset by a human-in-the-loop pipeline. Then, a cascaded pipeline based on SAM-H model and ResNet-18 was used for MF detection. Overall, the paper is well-written and the proposed modules in the pipeline are well-motivated. However, the clarification and experiments should be further improved to show the effectiveness of the proposed pipeline.

Abstract:

The Abstract should be written for the general audience. Please avoid using too many abbreviations.

Fig 2: ResNet-18 is relative small and old. Please compare to the SOTA classifiers (e.g., Transformer-based, ConvNeXt): <https://huggingface.co/spaces/timm/leaderboard>

Fig 3: Please show some typical examples where OMG-Net is better than previous SOTA.

Line 126: In this paragraph, please explain why Neuroendocrine tumor has lower performance than other cancer types.

Line 208: "Object detection models such as Faster R-CNN [30], RetinaNet [31] and YOLO [32] have been widely used for MF detection"

These methods [30-32] were proposed five years ago. Many new detection models have been proposed. Please test at least one latest detector to support your claims, e.g., RT-DETR

<https://github.com/lyuwenyu/RT-DETR>

Discussion:

Please introduce the potential direction to improve the MF detection performance for neuroendocrine tumor?

Please summarize the remaining challenges for MF detection (e.g., the top three challenges).

Code: Please polish the readme file to add detailed guidelines on the model training and inference commands.

Data: Since dataset is one of the most important contributions in this work, please provide a data download link in the manuscript

Version 1:

Reviewer comments:

Reviewer #1

(Remarks to the Author)

Thank the authors for the detailed response. It addressed some of my concerns. But I have a few remarks:

1. In Table 2, how is cancer detected from the morphological features? What mechanism is used to cluster cells for tumor identification? This process is also unclear from Figure 2.
2. As echoed in R#2, I'm curious about the intermediate test results for Segment Anything — specifically, how well does the segment mask perform? For ResNet-50, has the author explored any special mechanisms to address the feature scale limitations?

Reviewer #2

(Remarks to the Author)

Reviewer #3

(Remarks to the Author)

Thanks for the detailed response. All my concerns have been well addressed.

Version 2:

Reviewer comments:

Reviewer #1

(Remarks to the Author)

Thanks for the detailed response. All my concerns have been well addressed.

Reviewer #1

Overall comment:

This study introduces an AI-aided approach to improve the detection of mitotic figures (MFs), a crucial factor in cancer grading, in digitized haematoxylin and eosin-stained whole slide images. Recognizing MFs is traditionally time-consuming and prone to human error, which can lead to incorrect cancer grading and suboptimal treatment. To address these challenges, the researchers created the largest pan-cancer dataset of MFs, combining an in-house soft tissue tumor dataset with five open-source datasets, resulting in 74,620 MFs. They developed a two-stage framework, OMG-Net, which uses the Segment Anything Model (SAM) for cell contouring and an adapted ResNet18 for MF classification. OMG-Net achieved an F1-score of 0.84 in pan-cancer MF detection, significantly outperforming the previous state-of-the-art MIDOG++ model, especially in breast cancer detection, by 16%. This approach offers superior accuracy in detecting MFs across various tumor types and scanning methods.

1.1. The paper lacks an analysis of the disadvantages of current mitotic activity detection models.

1. Developing and validating pan-cancer mitotic figure detection models remains a significant challenge due to the absence of extensive pan-cancer mitotic figure datasets. **(Line 95 - 103)**
2. The mitotic index is key for soft tissue tumour diagnosis. However, no studies have been presented on mitotic figure detection in human soft tissue tumours. **(Line 103 - 110)**
3. Integrating the detection and classification can limit the overall performance of detection models. **((Line 262 - 297, and please also refer the reply to comment 2.7)**

1.2. The work primarily focuses on methodological aspects, with insufficient discussion on the motivation of reinforcement learning. More context on the specific topic is needed.

We aim to deploy reinforcement learning in our future study for validating the model in clinical settings by including pathologists to constantly correct the detection and give tumour type-specific and image-specific feedback to enhance the model performance. **(Line 392 - 394)**

1.3. The authors seem to have missed some relevant literature. Specifically, they don't discuss learning-based methods for image-level tasks, missing out on several relevant citations: "Class-aware adversarial transformers for medical image segmentation", "Momentum contrastive voxel-wise representation learning for semi-supervised volumetric medical image segmentation", "SimCVD: Simple

We thank the reviewer for bringing those inspiring papers to our attention. We did find that the papers discussing about fundamental concepts of deep learning for medical image analysis are relevant to our study. As a result, we cited "Incremental learning meets transfer learning: Application to multi-site prostate mri segmentation" **(Line 337)** and "Rethinking Semi-Supervised Medical Image

contrastive voxel-wise representation distillation for semi-supervised medical image segmentation", "Incremental learning meets transfer learning: Application to multi-site prostate mri segmentation", "Bootstrapping semi-supervised medical image segmentation with anatomical-aware contrastive distillation", "Unsupervised wasserstein distance guided domain adaptation for 3d multi-domain liver segmentation", "Mine your own anatomy: Revisiting medical image segmentation with extremely limited labels", "Rethinking Semi-Supervised Medical Image Segmentation: A Variance-Reduction Perspective", "ACTION++: Improving Semi-supervised Medical Image Segmentation with Adaptive Anatomical Contrast", and "Implicit Anatomical Rendering for Medical Image Segmentation with Stochastic Experts". These methods are relevant to the method proposed in this paper. These relevant papers should be included in the reference list.

Segmentation: A Variance-Reduction Perspective” (**Line 376**) in the discussion to propose some potential improvements in our future work.

However, our study specifically focused on detecting cells in mitosis from digitised pathology slides, an object detection tasks from large size 2D images. The aim of our study is automatically counting the number of mitotic figures to make tumour grading more reproducible. Although segmentation for nuclei is used in the model, this is not the focus in this paper. Therefore, we cannot cite the papers that are specific for semantic segmentation for anatomical structures. We appreciate the value of those papers while we find it will be more suitable to cite those papers when publishing our other studies related to auto-segmentation for radiotherapy planning in the future.

Overall comment:

Very interesting paper and good work in general.

2.1. Line 99: From what I understand, you used the pHH3 as a marker to generate a training dataset. How was the staining protocol done? From my understanding, if you first stain with H&E and then with the IHC, due to the quite heavy processing of the tissue to apply the H&E (heating up to a high degree) and the destaining process, it leaves the tissue quite altered. Also, pHH3 can detect MF but it might produce "false positives" due to the fact that it detects the mitotic process quite early even on cells that might not appear to be mitotic figures on the H&E. Did you have some classifying the pHH3 detections as well?

We admit that the slides could be altered due to the de-staining and re-staining process. In our study this impact was mediated by two approaches: (1) We first applied registration on both slide-level and patch-level to find the location of the mitotic figures stained by the pHH3-antibody in the H&E-stained slides; **(Line 431 - 433)** (2) We got the initial masks of the mitotic figures by thresholding the pHH3-staining for their nuclei, and then converted the masks to bounding boxes and deployed the Segment Anything Model to generate more precise masks using those bounding boxes as prompts. **(Line 465 – 467)**

It is agreed that pHH3 detects mitoses in early stage. In our study, both the H&E and pHH3 images of the nuclei detected were reviewed by pathologists as a part of the data generation protocol to make sure the cells included in the dataset are in mitosis. **(Line 433 - 435)**

2.2. Figure 2: When using the SAM to do the segmentation, how do you ensure that SAM will segment structures of cell level and not larger structures?

Objects were retained based on their areas, which ranged from 36 pixels to 3600 pixels ($2.25 \mu\text{m}^2$ to $225 \mu\text{m}^2$ in slides with pixel size of $0.25 \mu\text{m}/\text{pixel}$). Using this range, we excluded objects that are significantly larger or smaller than typical cells, while included both normal and atypical mitotic figures. **(Line 479)**

2.3. Line 127: You mention that you benchmarked against the current state-of-the-art MIDOG++ but the MIDOG++ is a dataset and the model presented in the paper is not state-of-the-art. It was used merely to demonstrate how the dataset varies and the detection performance when an off-the-shelf detector is used. The state-of-the-art would be the top-3 models on the leaderboard of the MIDOG grand challenge.

The MIDOG++ is a newer version published after the MIDOG 2022 challenge. Since it is the largest and the most diverse mitotic figure dataset to date, we decided to use it in our study.

The test set of the MIDOG 2022 challenge is not publicly available, for which we cannot compare our model to the models published in the final report of the challenge.

	However, we agree that comparing our model to the top models on the leader board is informative. Therefore, we included a new table in the manuscript (Table R2) to report the scores of different models on the subset included in both the test sets of MIDOG ++ and the MIDOG 2022 challenge, namely human melanoma, and canine cutaneous mast cell tumour. We also extended Table 1 (Table R1) to show the scores of our model on the full MIDOG ++ test set, please refer to the reply to the comment 2.9.
2.4. line 129: typo 'hows'  'shows'	It has been corrected in the manuscript.
2.5. Table 2: You chose to compare against a subset of the dataset using the leave-one-out(?)training results. Why was this choice made?	We compared the scores of our model to the results of the model trained using all the domains. In the MIDOG++ paper they included the results in the last row of Table 4 (https://www.nature.com/articles/s41597-023-02327-4/tables/4).
2.6. Line 146: You refer to "only RGB images" and "RGB classifier". What do you mean exactly? Is there a dataset of non-RGB images?	The “only RGB images” means the classification was done using the H&E-stained images of the cells solely; One of the highlights of our paper is that we found using a 4-channel input, which contains the binary masks of the nuclei in addition to the RGB images can improve the classification. (Figure 4)
2.7. Line 224: You mention that decoupling the object detection and classifier steps improves the performance because the existing models have to reject many false positives due to the imbalance of the dataset. However in your description you use SAM to generate masks and then apply a classifier. How does that approach exactly generate less false positives? If you were to segment all cells in a ROI, it would surely generate as many false positives as a detector I would assume.	In this section we claimed that the performance can be restricted due to the imbalance of the loss function, not the imbalance of the dataset. In most of the object detection models, the detection loss and classification loss are summed up to train the model. However, in real world settings the two parts of loss is not always equally important. For example, when detecting the mitotic figures, we will focus more on the classification accuracy rather than the quality of the detected bounding boxes or masks. Therefore, we do not want the two parts of loss to be penalised together as it is challenging to determine the ideal weights for each loss, and it is also possible that the linear combination of the losses is

	not optimal. This is why a lot of studies used a second-stage classifier to further reduce the number of false positives. Our two-stage framework completely separated the detection loss and classification loss. We deployed the Segment Anything Model pre-trained using a massive amount of data to provide high-quality masks, and used a large number of points as the prompts to make sure most of the mitotic figures will be included in the cells detected by the Segment Anything Model. The classification stage is independent to the detection stage, for which the classification scores was optimized fully independently. To summarise, in this session we claim that optimizing the classification loss independently can improve the overall performance for detecting mitotic figures.
2.8. You used for the classifier a ResNet-18, did you test other models as well?	Yes, we added a new table in the supplementary materials showing the classification performance of various image classifiers (Table R3). We compared the ResNet18 to ResNet50, DenseNet121, ConvNeXt as suggested by Reviewer 3, and EfficientNet-B7, the classifier used by the top 1 team on the MIDOG 2022 leader board. More detailed discussion of the comparison has been added to the reply to comment 3.2.
2.9. I would like to see the performance of this method using the same method as that was outlined during the grand-challenge. That means running in on the full test set, not just the human in order to have a direct comparison with the leaderboard of the challenge. By only comparing specific tumor types, it feels like cherry-picking the results. If the method is better, then it would perform better on the whole test dataset. After you have shown in direct comparison that the method is better, you can further evaluate with an extensive	The aim of this paper is automatically detecting mitotic figures for human cancer diagnosis. Therefore, we focused on reporting the performance on human subsets rather than the canine subsets as it makes the model selection more suitable for clinical application. However, we agree that reporting the scores on the whole testing set could be informative, for which we added the results for the whole testing set in the result section (Table R1) and highlight the performance for human tumours in the discussion.

training dataset.	As mentioned in the reply to comment 2.3 , we are not able to reproduce the exact same comparison using the test set of MIDOG 2022 challenge, we compared our model to the top 3 models from the challenge by comparing the test scores on the overlapped subsets of MIDOG++ and MIDOG 2022.
-------------------	---

Reviewer #3
Overall comment: This work first presents an annotated mitotic-figure (MF) dataset by a human-in-the-loop pipeline. Then, a cascaded pipeline based on SAM-H model and ResNet-18 was used for

MF detection. Overall, the paper is well-written and the proposed modules in the pipeline are well-motivated. However, the clarification and experiments should be further improved to show the effectiveness of the proposed pipeline.

3.1. Abstract: The Abstract should be written for the general audience. Please avoid using too many abbreviations.

The abstract has been updated:

Mitotic activity is an important feature for grading several cancer types. However, counting mitotic figures (cells in division) is a time-consuming and laborious task prone to inter-observer variation. Inaccurate recognition of MFs can lead to incorrect grading and hence potential suboptimal treatment. This study presents an artificial intelligence-based approach to detect mitotic figures in digitised whole-slide images stained with haematoxylin and eosin. Advances in this area are hampered by the small size and variety of datasets available. To address this, we have created the largest dataset of mitotic figures (N=74,620), combining an in-house dataset of soft tissue tumours with five open-source datasets. We then employed a two-stage framework, named the Optimized Mitoses Generator Network (OMG-Net), to identify mitotic figures. This framework first deploys the Segment Anything Model to automatically outline cells, followed by an adapted ResNet18 that distinguishes mitotic figures. OMG-Net achieved an F1 score of 0.84 in detecting pan-cancer mitotic figures, including human breast carcinoma, neuroendocrine tumours, and melanoma. It outperformed previous state-of-the-art models in hold-out test sets. To summarise, our study introduces a generalizable data creation and curation pipeline and a high-performance detection model, which can largely contribute to the field of computer-aided mitotic figure detection.

3.2. Fig 2: ResNet-18 is relatively small and old. Please compare to the SOTA classifiers (e.g., Transformer-based, ConvNeXt): https://huggingface.co/spaces/timm/leaderboard	Since the aim of using the classifier is to classify mitotic figures and other cells, the input size is relatively small (64 pixel × 64 pixel). Although more advanced larger models show better performance of classifying more complex and diverse images, they did not bring better performance on our task with small input size for binary classification. We compared the ResNet18 to ResNet50, DenseNet121, ConvNeXt and EfficientNet-B7, the classifier used by the top 1 team on the MIDOG 2022 leader board (Table R3).
3.3. Fig 3: Please show some typical examples where OMG-Net is better than previous SOTA.	Figure 3 has been updated with some examples of detection results.
3.4. Line 126: In this paragraph, please explain why Neuroendocrine tumour has lower performance than other cancer types.	 1. The number of annotations for neuroendocrine tumour is relatively small (524) in the training data. All the annotations were from the MIDOG++ and the images were scanned using the same scanner. The small amount of data and the lack of diversity makes it more challenge to develop robust and accurate models. 2. The poor performance can also be attributed to the biological characteristics of neuroendocrine tumours. Cell aggregation, where cells clump together, is often observed in neuroendocrine tumour specimens, for which it is difficult to distinguish individual cells' boundaries and limit the quality of the nuclei masks. The heterogeneity of aggregation also brings more diverse cell morphology, making it more challenging to recognize and classify them correctly. 3. In our dataset, the images of neuroendocrine tumours frequently suffer from poor quality, including issues like blurring, uneven staining, and artifacts. Low-quality images can lead to increased false negatives or false positives, as the model struggles to

	interpret the compromised visual information. The interpretation of the results for neuroendocrine tumour is added in Discussion (Line 344 - 370)
3.5. Line 208: “Object detection models such as Faster R-CNN [30], RetinaNet [31] and YOLO [32] have been widely used for MF detection” These methods [30-32] were proposed five years ago. Many new detection models have been proposed. Please test at least one latest detector to support your claims, e.g., RT-DETR https://github.com/lyuwenyu/RT-DETR	We retrained the RT-DETR model using our dataset and tested it on the same test set. The dataset was prepared as patches with size of 640 pixels × 640 pixels. A RT-DETR-X was trained for 200 epochs using the default configuration. Overall, we observed a lower F1 score comparing to OMG-Net (0.764 ± 0.01 vs 0.783 ± 0.02).
3.6. Discussion: Please introduce the potential direction to improve the MF detection performance for neuroendocrine tumour?	 1. Increasing the size of the training datasets. By incorporating a larger and more diverse set of annotated images of neuroendocrine tumours, the model will be accurate and robust on detection mitotic figures from neuroendocrine tumours. We will constantly collect specimens from RNOH and collaborative institutions and apply the data generation and curation pipeline introduced in this paper to expand the size and diversity of not only neuroendocrine tumours but also soft tissue tumours. 2. Improving the quality of the whole slide images used for training. This can be achieved through standardized protocols for slide preparation, staining and scanning. Meanwhile, we will implement quality control by pathologists before selecting images for developing the model. By ensuring the data quality, we can significantly enhance the detection performance. The discussion about the solutions is combined with the possible reasons in the Discussion (Line 344 - 370)

3.7. Please summarize the remaining challenges for MF detection (e.g., the top three challenges).

1. Detecting mitotic figures in rare tumours. Although a large number of studies reported datasets or detection models for mitotic figures in breast cancer, detecting mitotic figures in other tumour types, especially rare cancers remain challenging. The limited number of cases but high degree of heterogeneity of rare cancers such as soft tissue sarcoma, complicates the acquisition of ample and diverse data for training robust AI models. In this study, we established the largest-to-date mitotic figure dataset for human soft tissue tumour. We are committed to continuously expanding this dataset and refining our models to enhance their accuracy and utility in sarcoma diagnosis.
2. Comparing the mitotic index identified by AI and by pathologists. Clinically, the mitotic index is obtained by pathologists counting the number of mitotic figures within 10 high-power fields. However, this process is subject to intra- and inter-observer variability, which can significantly affect the consistency and reliability of the measurements, brining challenge to align manual counts to the mitotic index given by AI-driven whole-slide image analysis. Standardised protocol is needed for comprehensively assessing the accuracy and potential clinical impact of the AI mitotic index.
3. Clinical implementation of AI models for mitotic figure detection. While AI-based models for detecting mitotic figures have demonstrated exceptional performance in research settings, translating these advances into clinical practice remains a challenge. The key issues include allocating sufficient computational resources in clinic, ensuring

	compliance with clinical data regulation and receiving feedback and new data for constant model improvement. Our future work also includes designing federated learning platform to facilitate the model refining, which allows multiple institutions to expand the training data while keeping sensitive data localized. A whole section to discuss the remaining challenges is added in Discussion (Line 339 - 394).
3.8. Code: Please polish the readme file to add detailed guidelines on the model training and inference commands.	https://github.com/SZY1234567/OMG-Net
3.9. Data: Since dataset is one of the most important contributions in this work, please provide a data download link in the manuscript.	https://zenodo.org/records/11521640

Changed Tables & Figures

Table R1. Precision, recall and F1 scores in MIDOG++ testing set of OMG-Net against the model presented by MIDOG++.

Tumour Types	Precision	Recall	F1	Ensemble F1	F1(MIDOG++)
breast carcinoma	0.82 ± 0.02	0.88 ± 0.02	0.85 ± 0.02	0.87	0.71 ± 0.02
neuroendocrine tumour	0.64 ± 0.02	0.65 ± 0.03	0.64 ± 0.02	0.67	0.59 ± 0.02
melanoma	0.83 ± 0.02	0.84 ± 0.03	0.83 ± 0.01	0.85	0.81 ± 0.02
lung carcinoma*	0.69 ± 0.02	0.70 ± 0.02	0.69 ± 0.02	0.74	0.68 ± 0.02
lymphosarcoma*	0.76 ± 0.03	0.74 ± 0.01	0.76 ± 0.03	0.80	0.73 ± 0.01
cutaneous mast cell tumour*	0.84 ± 0.02	0.88 ± 0.02	0.86 ± 0.01	0.87	0.82 ± 0.01
soft tissue sarcoma*	0.74 ± 0.02	0.73 ± 0.02	0.74 ± 0.02	0.77	0.69 ± 0.01

*Canine Specimens.

Table R2. F1 scores of OMG-Net and the top 3 models from the MIDOG 2022 challenge in the overlap subsets between the MIDOG++ and MIDOG 2022 test sets.

Tumour Types	OMG-Net	TIA Centre	TCS Research	USZ/UZH Zurich
melanoma	0.85	0.80 [0.74,0.84]	0.76 [0.66,0.80]	0.79 [0.74,0.83]
cutaneous mast cell tumour*	0.87	0.83 [0.81,0.86]	0.76 [0.58,0.83]	0.73 [0.66,0.79]

*Canine Specimens.

Table R3. Classification metrics of different classifiers tested for mitotic figure classification.

Classifier	Precision	Recall	F1	Accuracy	AUC
ResNet18	0.851	0.842	0.846	0.995	0.949
ResNet50	0.842	0.836	0.838	0.995	0.928
DenseNet121	0.834	0.820	0.827	0.994	0.924
ConvNeXt-tiny	0.839	0.845	0.842	0.994	0.946
Efficient-Net-B7	0.841	0.840	0.841	0.994	0.941

Figure 1: Detection performance. **a** The testing F1 scores in the human subsets of the proposed framework, where the yellow dashed lines mark the ensemble F1 scores and the red dashed lines mark the mean F1 scores reported by MIDOG++. **b** The changes in the average F1 score as more mitotic figures (MFs) are included in training. **c** The detection results of the OMG-Net and the Retina-Net used in the MIDOG++ in example regions. The green, yellow and red bounding boxes represent the true positives, false positives and false negatives.